# Effects of Nutritional Education and Diet on Obesity and Asthma Progression in Children and Adolescents

**DOI:** 10.3390/nu17111759

**Published:** 2025-05-23

**Authors:** Monika Soczewka, Justyna Waśniowska, Bogda Skowrońska, Aleksandra Szczepankiewicz, Irena Wojsyk-Banaszak, Andrzej Kędzia, Elżbieta Niechciał

**Affiliations:** 1Department of Pediatric Diabetes, Auxology and Obesity, Poznan University of Medical Sciences, 60-572 Poznan, Poland; monikasoczewka22@gmail.com (M.S.); akedzia@ump.edu.pl (A.K.); 2Doctoral School, Poznan University of Medical Sciences, 60-812 Poznan, Poland; 3Department of Human Nutrition and Dietetics, Faculty of Food Technology, University of Agriculture in Krakow, 31-120 Krakow, Poland; justyna.wasniowska@student.urk.edu.pl; 4Department of Pulmonology, Pediatric Allergy and Clinical Immunology, Poznan University of Medical Sciences, 60-567 Poznan, Poland; alszczep@gmail.com (A.S.); iwojsyk@ump.edu.pl (I.W.-B.)

**Keywords:** children, adolescents, obesity, asthma, nutrition, anti-inflammatory diet, nutritional education

## Abstract

**Background**: Asthma and obesity commonly co-occur in children, with obesity contributing to asthma development through inflammatory and mechanical pathways. A Mediterranean diet may reduce inflammation and improve outcomes. However, research on the effects of nutritional interventions and dietary education in children with asthma and obesity within the Polish population remains limited. **Methods**: 142 participants were enrolled in the observational study and divided into three groups: children with asthma and overweight/obesity, children with overweight/obesity, and a control group. Anthropometric and clinical data, dietary habits, and lifestyle parameters (sleep duration, physical activity, screen time) were assessed at baseline and after one year of nutritional intervention. The Mediterranean diet was the main dietary model advocated throughout the intervention. Diet quality and adherence to the Mediterranean dietary pattern were evaluated using the KIDMED 2.0 index, reflecting the anti-inflammatory components of the intervention. Nutritional education was delivered every 6–8 weeks. One-way analysis of variance (ANOVA) was used, with significance set at *p* ≤ 0.05. **Results**: The nutritional intervention led to significant improvements in metabolic parameters, evidenced by reductions in total cholesterol, triglycerides, and fasting glucose levels. A decrease in C-reactive protein levels indicated reduced inflammation. The adherence index to the Mediterranean diet, measured by the KIDMED 2.0 scale, significantly increased in all groups, with the most favorable effects observed in the obese/overweight group with asthma. Positive behavioral changes were also noted, including increased physical activity, longer sleep duration, and reduced screen time. The intervention also contributed to a significant improvement in participants’ nutritional knowledge. **Conclusions:** Dietary education and an anti-inflammatory diet improved health in children with asthma and obesity. Early nutritional interventions helped reduce inflammation, improve metabolism, and shape behaviors. The findings stress the need to integrate diet education into therapeutic and preventive strategies for affected pediatric populations.

## 1. Introduction

Childhood obesity and asthma are prevalent and often coexisting conditions in pediatric populations worldwide, including Poland. Despite the growing interest in dietary approaches to managing these conditions, there is still a lack of studies assessing the impact of diet and nutrition education on asthma and obesity in Polish children and adolescents [1,2]. This research gap underscores the need for targeted, evidence-based strategies tailored to national dietary habits and health profiles. Obesity contributes to the development and severity of asthma through both mechanical and inflammatory mechanisms. Adipose tissue secretes pro-inflammatory cytokines, triggering low-grade systemic inflammation and exacerbating asthma symptoms [1]. Evidence suggests that appropriate nutrition, particularly anti-inflammatory dietary patterns, can help alleviate these processes [2]. One of the best-studied anti-inflammatory dietary models is the Mediterranean diet (MD), characterized by a high intake of vegetables, fruits, whole grains, legumes, and healthy fats such as olive oil. Moderate consumption of fish and poultry supports dietary balance. This pattern has been associated with reduced inflammation and improved respiratory outcomes [1,3,4,5]. In contrast, the Western diet—high in processed foods, saturated fats, and added sugars—has been linked to worsened health parameters and may aggravate asthma symptoms [2,4].

Studies show that the daily diets of school-aged children are often dominated by calorie-dense, nutrient-poor foods, with insufficient intake of fruits and vegetables [6]. Nutrition education can play a key role in reversing these patterns, especially in children affected by both asthma and obesity. Educational interventions that promote the consumption of fiber and antioxidant-rich foods, along with physical activity, have been shown to improve body mass index (BMI), lung function, and asthma control [1,7]. Longer-term programs, lasting at least six months, can lead to significant changes in eating behaviors and health outcomes, supporting weight reduction and improving asthma-related parameters [8].

The present study aims to evaluate the effects of a structured dietary intervention and nutrition education on asthma and obesity in the pediatric population in Poland. The findings may serve as a foundation for developing effective national strategies for preventing and managing these conditions [2].

## 2. Materials and Methods

### 2.1. Study Design and Ethics Issue

This observational study was conducted according to the Strengthening the Reporting of Observational Studies in Epidemiology [9] and the Declaration of Helsinki [10]. The study protocol was approved by the Ethical Committee of the Poznan University of Medical Sciences (protocol no. 819/21, date of approval: 4 November 2021, with amendments). All participants’ parents and participants > 16 years old provided written informed consent before enrolment.

### 2.2. Minimum Sample Size Calculation

The minimum sample size was calculated as 42 subjects per group using the G*Power 3.1 software (University of Kiel, Kiel, Germany) to obtain a power of 80% (Cohen’s d = 0.5, α = 0.05, β = 0.2). Considering a maximum dropout rate of 20%, each group should contain at least 50 subjects.

### 2.3. Study Population

Participants were recruited from July 2021 to December 2023 at the Department of Pediatric Diabetes, Auxology, and Obesity, Poznan University of Medical Sciences; the Department of Pneumonology, Allergology, Pediatric and Clinical Immunology, Poznan University of Medical Sciences; the Diabetology Outpatient Clinic; and the Online Clinic dealing with dietary therapy of obesity. One hundred ninety-one children were recruited, but forty-nine participants withdrew during the survey without giving a reason. There were three research groups: 1—obese/overweight group without asthma (O)—children aged over 6 years, with BMI > 85th percentile (overweight), >97th percentile (obese) for age and sex (exclusion criteria: immunotherapy in the last 6 months, lack of consent to participate in the study); 2—obese/overweight group with asthma (AO): children with asthma aged over 6 years with BMI > 85th percentile (overweight), >97th percentile (obese) for age and sex, asthma—physician’s diagnosis, antiasthmatic treatment (e.g., inhaled glucocortoicosteroids albo clinical symptoms (at least 2 of the following: cough, wheezing, dyspnoe, impaired exercise tolerance) (exclusion criteria: immunotherapy in the last 6 months, systemic steroids in the previous 4 weeks, comorbidities (immunodeficiencies, autoimmune diseases), lack of consent to participate in the study); 3—control group—exclusion of allergy and asthma (C) aged over 6 years with normal BMI (between the 15th–85th percentile for sex and age).

### 2.4. Data Collection Timeline

The education intervention process is summarized in Table 1. At the first visit, laboratory results, including fasting glucose and insulin levels, glycated hemoglobin, and lipid profile: total cholesterol, high-density lipoprotein, low-density lipoprotein, triglycerides, and C-reactive protein level, were provided by patients based on analyses performed at certified diagnostic laboratories. Participants were also given dietary questionnaires and a 3-day food diary to complete at home for submission during their next visit to a qualified nutritionist [11].

**Table 1 nutrients-17-01759-t001:** The education intervention process.

Stage	Time Point	Participants	Activity/Assessment	Purpose
1	Baseline (Visit 1)	All groups (AO, O, C)	Anthropometrics, fasting labs, food diary, YHEI, KIDMED 2.0, VAS	Initial health and dietary status assessment
2	Week 0	AO and O	Individualized nutrition consultation (1)	Interview, goal setting, and motivation
3	Every 6–8 weeks (10 sessions total)	AO and O	Structured educational sessions (see Table 2 in manuscript) + follow-up, adaptation of recommendations	Gradual implementation of Mediterranean and anti-inflammatory dietary patterns
4	Throughout 12 months	AO and O	Motivational support, food diary analysis, and behavior reinforcement	Improve adherence, self-efficacy, and parental support
5	Month 12 (Final Visit)	All groups (AO, O, C)	Re-assessment: anthropometrics, labs, food diary, YHEI, KIDMED, VAS	Evaluation of intervention effectiveness

AO—Asthma and Overweight/Obesity group, O—Overweight/Obesity group, C—Control group; YHEI—Young Healthy Eating Index, KIDMED 2.0—Mediterranean Diet Quality Index; VAS—Visual Analogue Scale (knowledge and motivation assessment). The C group was not included in the educational intervention process.

**Table 2 nutrients-17-01759-t002:** Authors’ Education Program.

Consultation Number	Main Aim
1 consultation	Interview, introduction, setting individual nutrition and health goals. Motivation.
2 consultation	The basics of healthy eating are a plate of healthy food. Portion control, food awareness.
3 consultation	Anti-inflammatory diet—what to eat, what to avoid. Products with anti-inflammatory effects reduce inflammation. Glycemic index and glycemic load.
4 consultation	Role of fiber, fats, protein, and carbohydrates.
5–6 consultation	Meal planning, shopping, and reading food labels. The role of snacks, sugar, and sweeteners.
7–8 consultation	Practical tips for parents. The role of hydration, physical activity, and sleep.
9–10 consultation	Summary and consolidation of habits.

During all consultations, the presence of products with anti-inflammatory effects in meals was checked, and the previously communicated knowledge was consolidated, focusing mainly on elements of an anti-inflammatory diet.

### 2.5. Outcome Measures

Patients were weighed and measured according to standard measurement procedures using a calibrated scale and stadiometer, with the principles of accuracy and repeatability of measurements. All measurements were taken in underwear and after fasting in the morning. Height was assessed to the nearest 1 cm, while body weight was measured to the nearest 0.1 kg. BMI ± SD was calculated to determine the nutritional status of the study population according to the criteria [12,13].

Nutritional habits were evaluated using diary records. Participants recorded their diet in the diary for three days within one week, including one weekend day, which could be consecutive or non-consecutive. Based on the food diaries, the qualified dietitian assessed diet quality by the Young Healthy Eating Index (YHEI), completed the Nutrition Education’s Impact on Mediterranean Diet Quality Index (KIDMED 2.0), and also assessed their level of knowledge and motivation using the visual analogue scale (VAS) [14,15]. In addition, those with asthma and obesity were given the author’s Asthma Control (AC) questionnaire (Appendix A). Patients gave test results to the dietitian during the visit.

### 2.6. Intervention Description

Each patient, for one year, had an individual consultation with a qualified nutritionist every 6–8 weeks (stationary or online). During the visits, the dietitian provided nutritional education focused on the patient’s anti-inflammatory diet (Table 2). In addition to structured nutrition education, participants in the AO and O groups received individualized dietary recommendations tailored by a qualified dietitian. These recommendations were based on the anti-inflammatory diet principles and the Mediterranean dietary pattern. Rather than fixed meal plans, the guidance focused on practical suggestions for gradually improving nutritional habits, such as increasing the intake of vegetables, fruits, and healthy fats, and reducing processed foods. Recommendations were adjusted to individual preferences and assessed regularly during consultations every 6–8 weeks. After one year, patients again completed the food diary and the authors’ questionnaire and provided results of repeat laboratory tests and the measurements. A qualified dietitian assessed diet quality again by the YHEI and completed the KIDMED and level of knowledge and motivation.

### 2.7. Statistical Analysis

Data are presented as mean ± SD. One-way analysis of variance (ANOVA) was used to test for differences at *p* ≤ 0.05. Statistical analysis was performed using Statistica 13.1 PL. The Duncan test was used to test for differences between researcher groups. Importantly, the statistical analysis was carried out by another author, who was blinded to group allocation and had no contact with participants, helping to reduce potential observer bias during data interpretation.

## 3. Results

The study population was divided into three groups: Asthma and Obesity (AO), Obesity (O), and Control (C), with the AO and O groups consisting of 50 participants and the C group composed of 42. The group of children with asthma and overweight or obesity (AO) included 34 girls (F) and 16 boys (M), with a mean age of 12.54 ± 3.58 years and a mean Body Mass Index ± Standard Deviation (BMI ± SD) of 25.86 ± 4.16 kg/m^2^. The group of children with overweight or obesity (O) consisted of 23 girls and 27 boys, with a mean age of 11.02 ± 3.23 years and a mean BMI ± SD of 26.42 ± 4.86 kg/m^2^. The control group (C) included 24 girls and 18 boys, with a mean age of 12.24 ± 3.47 years and a mean BMI ± SD of 18.07 ± 1.66 kg/m^2^. The characteristics of the study population are shown in Table 3.

Concerning the lipidogram, statistically significant differences were noted between the AO-before and AO-after and O-before and O-after groups, respectively. Children following elements of the anti-inflammatory diet (AO-after, O-after) had lower serum TC levels compared to those before the diet (AO-before, O-before). At the same time, HDL levels increased significantly in the AO and O groups. LDL and TG values also decreased in both study groups after the intervention, but a significant decrease in LDL occurred in the O group and TG in the AO group (Table 4).

Regarding glycemia, fasting glucose was significantly decreased in both groups (AO and O). The decrease in insulin in both groups (AO, O) was not statistically significant. There was a decrease in HbA1c and homeostatic model assessment for insulin resistance (HOMA-IR) post-intervention, significantly only in the AO group (Table 4).

In the study group of children AO, the effect of dietary intervention on selected clinical and laboratory indicators and patients’ subjective feelings, measured before (AO-before) and after the intervention (AO-after), was evaluated. The NDAS parameter (average number of days of asthma symptoms per month) decreased significantly. Similarly, the levels of C-reactive protein (CRP) (*p* < 0.05) were reduced statistically considerably (Table 5).

The YHEI index values analysis showed significant differences between the study groups before and after the intervention. After the intervention, the AO and O groups observed an increase in the YHEI index. After the intervention, a significant increase in the YHEI index was observed only in group O (*p* < 0.05). This group also had the highest YHEI scores at both baseline and post-intervention. In contrast, the YHEI index in the control group remained unchanged (Figure 1a).

In the control group (C), the mean sleep length before and after the intervention did not change significantly. In the AO and O groups, the mean sleep length before the intervention was significantly shorter compared to the C group. After the intervention, a significant increase in mean sleep length was observed in both groups (AO and O) compared to pre-intervention parameters (Figure 1b).

After implementing education, the time spent in front of the screen significantly changed in the AO and O groups. A significant decrease in time was observed in both groups. In group C, no significant differences existed before and after the intervention was applied (Figure 1c).

In children with asthma and obesity (AO), the level of physical activity before the intervention was the lowest among all groups. In the AO and O groups, there was a significant increase in the time devoted to physical activity. In group C, no significant differences existed before and after the intervention was applied (Figure 1d).

There was an increase in motivation in all groups (C, AO, O). However, the AO group showed a significant statistical increase in motivation after the intervention (Figure 1e). 

Before the intervention, the level of knowledge was lowest in the AO group (Figure 1f).

Figure 2a presents the absolute values of KIDMED 2.0 scores before and after the intervention in each group, while Figure 2b illustrates the percentage increase in the KIDMED score across the groups. All study groups (C, AO, O) showed a significant statistical increase in knowledge about the diet after the intervention. The groups were statistically significantly different from each other. The AO group had the highest post-intervention scores on the KIDMED 2.0 index (Figure 2a). The C group had the highest percentage increase of 85.5%, followed by the AO group at 63.5% and the O group at 25% (Figure 2b).

Figure 3a shows the raw scores from the Asthma Control (AC) questionnaire before and after the intervention in the AO group, while Figure 3b presents the percentage improvement in asthma control. The results obtained from the AC assessment after implementing the nutritional intervention showed a decrease in the score obtained (Figure 3a). The results indicate a 20.42% improvement in asthma control after the intervention (Figure 3b).

## 4. Discussion

Chronic inflammation plays a critical role in developing many non-communicable diseases, highlighting the importance of early dietary interventions in childhood. The findings of this study suggest that a comprehensive nutritional approach—consisting of education and the promotion of an anti-inflammatory diet—may contribute to improved metabolic outcomes and better asthma control in children with overweight or obesity.

Currently, the MD is recognized as a valuable dietary intervention. It is characterized by a high intake of vegetables, fruits, nuts, whole grains, and olive oil; moderate consumption of fish and poultry; and limited consumption of sweets, red meat, and dairy products [17]. Previous studies indicate the beneficial effects of MD in improving glycemic control, regulating blood pressure, and preventing cardiovascular disease by reducing inflammation [6,18,19,20]. In addition, this diet supports the immune system. It reduces inflammatory mediators, resulting in an overall improvement in health, as indicated by the results of our study, which we refer to later in the discussion.

Analysis of the lipid profiles of the participants showed a significant improvement. TC levels decreased in the group of children with asthma and obesity and the group of children with obesity. TGs also decreased in both study groups. These observations are consistent with the findings of Rodríguez-Cano et al. [21], which confirm the effectiveness of dietary interventions in reducing dyslipidemia in children with obesity. Metabolic changes were also evident in glucose and insulin levels. HbA1c values decreased in the AO and O groups, as did the HOMA-IR index, indicating improved insulin sensitivity. These results are consistent with the study by Gobato et al. [22], which suggests that appropriate dietary modifications can improve glycemic parameters and reduce insulin resistance in overweight and obese children; significant changes were also observed in inflammatory markers such as CRP. The authors’ study also indicates that MD is associated with substantial parameter improvements. As indicated by previous studies [23,24,25,26], a significant reduction in the number of days with asthma symptoms was observed in our analysis. The diet also reduced inflammation, as reflected by lower levels of CRP. These results are consistent with the study by Romieu et al., which indicated a beneficial effect of the MD diet on lung function in children with asthma [27]. In contrast, Calatayud-Sáez et al. noted that after a year of following the MD, the severity of infections, the number of asthma attacks, hospitalizations, and the need for medications decreased significantly [28]. Our results suggest that adherence to an MD may be associated with reduced asthma symptoms and fewer days with respiratory complaints, which could support its role as a complementary approach in managing asthma in children with obesity. Jensen ME. et al. also reported a positive effect of dietary intervention on improving asthma control [29]. Although the observed changes in metabolic parameters, such as total cholesterol and fasting glucose, reached statistical significance, their absolute magnitude was modest. Therefore, the immediate clinical relevance of these changes may be limited. However, even minor but directionally consistent improvements may hold potential importance in the context of long-term nutritional interventions, particularly in pediatric populations with obesity and chronic conditions. These findings may serve as a foundation for future studies with extended follow-up periods and targeted clinical endpoints.

The results of this study indicate that intensive diet and nutrition education effectively improve adherence to the MD. Similar results were obtained by Korkmaz et al. in a survey of 900 individuals with obesity and abdominal obesity from Ordu province. In their analysis, 45.6% of the children adhered to the diet moderately, 18.7% showed high levels of adherence, and 35.7% had low levels of MD adherence [30]. In contrast, a study of children attending elementary schools in northern Italy also found similar results. Most participants adhered to the diet at moderate levels, with 20.7% having low adherence, 60.3% having moderate adherence, and 19% having high adherence [31]. Another prospective cohort study among Spanish children found that 40.4% of children adhered to dietary recommendations at a high level, 50% at a moderate level, and 9.6% at a low level [32]. We also determined YHEI at the beginning of our study. After reassessment, YHEI values increased significantly in each study group after one year, indicating the inclusion of MD elements in daily nutrition and the effectiveness of nutrition education.

However, diet is only one component of a comprehensive lifestyle that affects children’s health. Behavioral and environmental habits, including time spent in front of a screen and physical activity levels, are essential aspects that require attention. Time spent in front of a screen refers to when people use electronic devices equipped with screens [33]. Meta-analyses have indicated that excessive use of such devices increases the likelihood of overweight or obesity in children, especially when the time exceeds 2 h per day [34,35]. In addition, prolonged use of screens can lead to sleep deprivation, reduced physical activity, and negative consequences for physical health [36]. Our study showed that children’s knowledge and motivation to make healthier choices increased after a year of educational intervention. This was evident in reduced time spent in front of a screen, improved sleep quality, and a greater desire to be active. These changes in behavior show that nutrition education can be an effective tool to support healthy lifestyles in children, further reinforcing the effects of dietary interventions.

## 5. Conclusions

The results of our study suggest that nutrition education may contribute to adopting an anti-inflammatory dietary pattern and be associated with improvements in specific health parameters among children with obesity and asthma. While reductions in metabolic markers and asthma-related symptoms were observed, these findings should be interpreted cautiously due to subjective measures and the absence of objective pulmonary function testing. Nonetheless, the study highlights the potential value of incorporating nutrition education into multidisciplinary strategies to improve pediatric health outcomes. Further studies using objective clinical endpoints and randomized designs are warranted.

## 6. Limitations of the Study

The study employed a non-randomized sampling strategy, which may have introduced selection bias and limited the generalizability of the results. The study did not include O, AO, or C groups without education. The data, including dietary intake, asthma symptoms, and behavioral changes, relied on self-reported information, subject to recall and social desirability biases. The form of consultation (online vs. in-person) varied among participants based on individual preferences and places of residence. While face-to-face contact with patients carries additional benefits, such as better behavioral observation and relationship building, the study’s findings indicate that online consultations can also be effective and yield real results. In today’s challenges of limited access to specialists in medical facilities, especially in small towns and among children with chronic diseases, the possibility of providing effective nutrition education remotely is a valuable alternative that can increase the availability and continuity of dietary care. However, the heterogeneity in consultation modes (online vs. in-person) could have influenced engagement levels and the effectiveness of education, although both formats followed a standardized protocol. Future research could explore whether the mode of education affects outcomes such as adherence, engagement, or behavioral change.

Additionally, we did not perform sex-specific analyses due to limited statistical power when stratifying participants by sex within already divided study groups. As sex may influence dietary intake, inflammatory markers, and asthma-related outcomes, this is a relevant limitation. Future studies should ensure sufficient sample sizes for subgroup analyses by sex.

Finally, due to the study’s exploratory nature, no formal correction for multiple comparisons (e.g., Bonferroni adjustment) was applied, which may increase the risk of false-positive results. Thus, the findings should be interpreted with appropriate caution.

## Figures and Tables

**Figure 1 nutrients-17-01759-f001:**
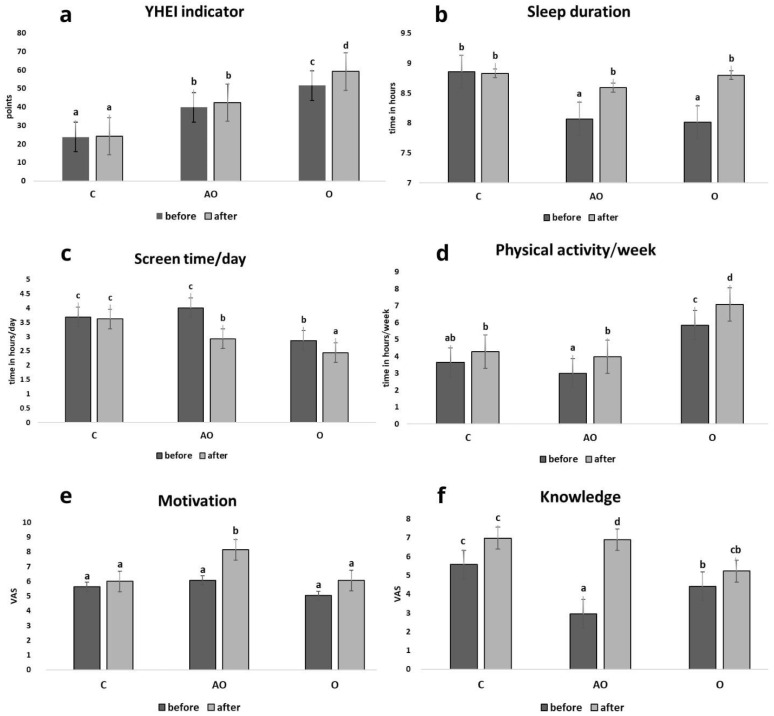
Selected lifestyle and behavioral indicators in all study groups. Comparison of lifestyle choices, behavioral indicators and motivation, knowledge before and after (**a**–**f**) the intervention in the control (C), asthma and overweight/obesity (AO), and overweight/obesity (O) groups. Values in rows with different letters (a, b, c, d) are significantly different, *p* ≤ 0.05 (One-way analysis of variance (ANOVA)).

**Figure 2 nutrients-17-01759-f002:**
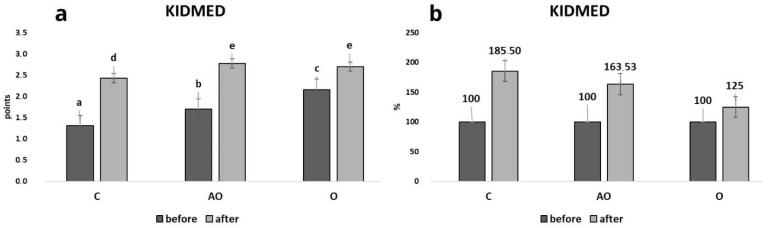
Mediterranean Diet Quality Index scores in all study groups. Changes in Mediterranean Diet Quality Index (KIDMED) (**a**,**b**) scores before and after the intervention in the control (C), asthma and overweight/obesity (AO), and overweight/obesity (O) groups. (**a**) Absolute KIDMED 2.0 scores before and after the intervention. (**b**) Percentage increase in KIDMED 2.0 scores. Values in rows with different letters (a, b, c, d, e) are significantly different, *p* ≤ 0.05 (One-way analysis of variance (ANOVA)).

**Figure 3 nutrients-17-01759-f003:**
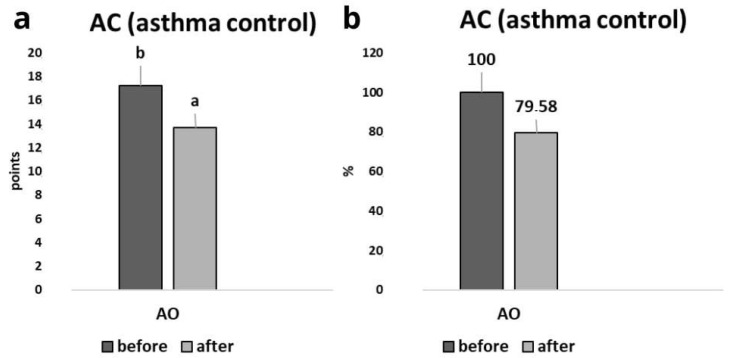
The author’s Asthma Control (AC) questionnaire (Appendix A). Changes in Asthma Control (AC) (**a**,**b**) scores in the asthma and overweight/obesity (AO) group before and after the intervention. (**a**) Raw AC scores before and after the intervention in the AO group. (**b**) Percentage improvement in asthma control. Values in rows with different letters (a, b) are significantly different, *p* ≤ 0.05 (One-way analysis of variance (ANOVA)).

**Table 3 nutrients-17-01759-t003:** Characteristics of the study group.

Group Name	N	Gender	Mean Age at Baseline [Years]	Mean BMI ± SD at Baseline [kg/m^2^]	Mean BMI BMI ± SD at the End of the Study [kg/m^2^]
Asthma and overweight/obesity	50	34 F	12.54 ± 3.58	25.86 ± 4.16	22.08 ± 3.30
		16 M			
Overweight/obesity	50	23 F	11.02 ± 3.23	26.42 ± 4.86	22.29 ± 4.45
		27 M			
Control	42	24 F	12.24 ± 3.47	18.07 ± 1.66	18.66 ± 1.89
		18 M			

**Table 4 nutrients-17-01759-t004:** Selected metabolic and health parameters were measured in all study groups.

	C-Before	C-After	AO-Before	AO-After	O-Before	O-After
Selected laboratory parameter	*n* = 42	*n* = 42	*n* = 50	*n* = 50	*n* = 50	*n* = 50
TC [mg/dL]	155.58 ^a^ ± 3.02	155.22 ^a^ ± 3.28	183.57 ^c^ ± 3.68	163.93 ^b^ ± 3.84	195.54 ^d^ ± 2.08	175.87 ^c^ ± 1.67
HDL [mg/dL]	45.51 ^abc^ ± 0.95	45.59 ^abc^ ± 1.18	42.50 ^a^ ± 1.29	49.49 ^c^ ± 1.16	43.43 ^ab^ ± 1.13	48.08 ^bc^ ± 1.09
LDL [mg/dL]	88.72 ^a^ ± 2.49	81.47 ^a^ ± 1.97	88.04 ^c^ ± 2.52	79.30 ^bc^ ± 2.39	99.00 ^c^ ± 2.08	87.35 ^b^ ± 2.01
TG [mg/dL]	85.69 ^a^ ± 4.02	82.60 ^a^ ± 3.50	119.58 ^d^ ± 3.74	99.03 ^c^ ± 2.85	99.89 ^bc^ ± 3.53	85.79 ^b^ ± 2.41
glucose [mg/dL]	87.84 ^b^ ± 1.38	87.39 ^b^ ± 1.46	95.02 ^c^ ± 1.74	89.95 ^b^ ± 1.84	86.84 ^b^ ± 1.65	81.92 ^a^ ± 1.62
insulin [μU/mL]	8.82 ^a^ ± 0.55	8.85 ^a^ ± 0.51	9.76 ^ab^ ± 0.82	8.81 ^a^ ± 0.80	12.87 ^c^ ± 0.97	11.57 ^bc^ ± 0.89
HbA1c [%]	5.35 ^ab^ ± 0.06	5.36 ^ab^ ± 0.06	5.50 ^b^ ± 0.07	5.26 ^a^ ± 0.07	5.73 ^c^ ± 0.07	5.41 ^ab^ ± 0.08
HOMA-IR	1.90 ^a^ ± 0.12	1.96 ^a^ ± 0.12	3.35 ^c^ ± 0.17	2.87 ^b^ ± 0.18	3.32 ^bc^ ± 0.16	2.93 ^bc^ ± 0.14

Selected metabolic and health parameters in three groups: C (control), AO (asthma and overweight/obesity), and O (overweight/obesity). Values in rows with different letters (a, b, c) are significantly different, *p* ≤ 0.05 (One-way analysis (ANOVA), standard error (*n* = 8)). ^a^ Values with the letter “a” do not differ significantly from each other. ^b^ Values with the letter “b” are significantly different from those with the letter “a” but do not differ from other values labeled “b”. ^c^ Values with the letter “c” are significantly different from those with both “a” and “b”. TC, total cholesterol; HDL, high-density lipoprotein; LDL, low-density lipoprotein; TG, triglyceride; HbA1C, glycated hemoglobin; HOMA-IR, Homeostasis Model Assessment of Insulin Resistance [16].

**Table 5 nutrients-17-01759-t005:** Asthma control parameters.

	AO-Before	AO-After
	*n* = 50	*n* = 50
NDAS (days/month)	15.26 ^b^ ± 1.23	7.40 ^a^ ± 0.97
CRP level (mg/dL)	1.46 ^b^ ± 0.07	1.01 ^a^ ± 0.09

Selected asthma control parameters in two groups: asthma and overweight/obesity before (AO-before) and overweight/obesity after (AO-after) the intervention. Values in rows with different letters (^a^, ^b^) are significantly different, *p* ≤ 0.05 (one-way analysis (ANOVA), standard error (*n* = 8)). ^a^ Values with the letter “a” do not differ significantly. ^b^ Values with the letter “b” are significantly different from those with the letter “a” but do not differ from other values labeled “b”. NDAS, average number of days with asthma symptoms; CRP level—C-reactive protein level.

## Data Availability

Data available in a publicly accessible repository.

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
