# Peer review of "Effects of Nutritional Education and Diet on Obesity and Asthma Progression in Children and Adolescents"

_nutrients, 2025, doi:10.3390/nu17111759_

Round 1
Reviewer 1 Report
Comments and Suggestions for Authors
Dear Authors,
Thank you for the opportunity to review your manuscript entitled "The effects of nutritional education and diet on obesity and the course of asthma in children and adolescents with obesity." The study addresses a highly relevant public health issue and provides valuable preliminary evidence for the benefits of dietary interventions in pediatric asthma and obesity. However, several sections require clarification or strengthening to enhance the overall scientific quality.
Below, I provide detailed comments structured by sections, including strengths and areas needing improvement.
- Introduction
Strengths:
- Provides a comprehensive background on the relationship between obesity, asthma, and diet.
- Cites a good number of recent and relevant references.
Improvements Needed:
- The research gap could be framed more sharply. Currently, the need for the study in the Polish population is mentioned late; it would strengthen the manuscript to state this clearly at the beginning.
- Consider summarizing the background more concisely to lead more quickly to the study objectives.
- Research Design and Methods
Strengths:
- The observational study design is appropriate for the research question.
- Clear ethical approval and sample size calculation are presented.
- Nutritional education intervention is described in considerable detail.
Improvements Needed:
- The Methods section is dense and challenging to navigate. Please consider using additional subheadings (e.g., "Intervention Description," "Outcome Measures," "Data Collection Timeline").
- The use of subjective reporting tools (dietary diaries, asthma symptom reports) introduces potential bias. Please discuss this more thoroughly under limitations.
- No mention of methods to minimize bias (e.g., blinding of data evaluators) — this should be acknowledged.
- Results
Strengths:
- Results are presented with appropriate statistical analysis (ANOVA), and group comparisons are clear.
- Behavioral outcomes (screen time, physical activity, sleep) are well integrated with clinical outcomes.
Improvements Needed:
- Although results are statistically significant, the magnitude of changes (especially metabolic parameters like cholesterol or fasting glucose) is modest. Please discuss the clinical relevance of these findings.
- Multiple statistical comparisons were performed without correction for multiple testing (e.g., Bonferroni adjustment) — the risk of false positives should be acknowledged.
- Discussion and Conclusions
Strengths:
- Interpretation connects well to the existing literature and reinforces the study's clinical importance.
- Highlights the potential practical benefits of online dietary consultations.
Improvements Needed:
- The discussion tends to overstate the impact of the intervention (particularly in asthma control), considering the limitations of subjective reporting and lack of objective lung function tests. Please moderate the conclusions accordingly.
- Limitations related to selection bias (non-randomized sample), reliance on self-reports, and heterogeneity of consultation mode (in-person vs. online) should be discussed more critically.
- Language and Style
Strengths:
- Generally understandable, and ideas are communicated.
Improvements Needed:
- The English language can be benefited by professional editing to improve flow and clarity. Many sentences are unnecessarily long and could be broken down for better comprehension.
- Minor translation oversights (e.g., "albo") should be corrected.
General Recommendations
- Add a stronger paragraph on the limitations of the study, focusing on subjective measures, self-report biases, and the potential impact of online consultations.
- Refine the conclusions to align more conservatively with the actual strength of the evidence presented.
- Perform a language review to enhance fluency, reduce redundancy, and correct minor grammatical errors.
- Optionally, consider presenting the education intervention process in a schematic table or flowchart to enhance clarity.
Overall Impression
Your study contributes valuable insights into the role of diet and education in managing obesity-related asthma among children and adolescents. With substantive revision, particularly regarding the discussion of bias and language refinement, the manuscript could make a strong contribution to the field.
Thank you again for your work on this important topic.
Comments on the Quality of English Language
Language and Style
Strengths:
- Generally understandable, and the ideas are communicated.
Improvements Needed:
- The English language can be benefited by professional editing to improve flow and clarity. Many sentences are unnecessarily long and could be broken down for better comprehension.
- Minor translation oversights (e.g., "albo") should be corrected.
Author Response
Response to Reviewer 1:
Thank you for the time and effort in reviewing and providing feedback on our manuscript, and we are grateful for the insightful comments and valuable improvements to our paper. Below, we provide the point-by-point responses.
Major comments:
- The research gap could be framed more sharply. Currently, the need for the study in the Polish population is mentioned late; it would strengthen the manuscript to state this clearly at the beginning. Consider summarizing the background more concisely to lead more quickly to the study objectives.
Answer: Thank you for this valuable comment. We revised the introduction to emphasize the research gap in the Polish pediatric population at the very beginning and streamlined the background section to lead more directly to the study objectives. Please see lines [44-73].
- The Methods section is dense and challenging to navigate. Please consider using additional subheadings (e.g., "Intervention Description," "Outcome Measures," "Data Collection Timeline").
Answer: Thank you for your helpful suggestion. We have revised the methods section by adding subheadings to improve clarity. Please see lines 75-162.
- The use of subjective reporting tools (dietary diaries, asthma symptom reports) introduces potential bias. Please discuss this more thoroughly under limitations.
Answer: Thank you for this valuable observation. We agree that the use of self-reported instruments, such as food diaries and asthma symptom questionnaires, introduces potential recall and social desirability biases. We have addressed this issue in more detail in the revised Limitations section, acknowledging the inherent constraints of subjective assessment tools, particularly in pediatric populations. Please see lines 349-364.
- No mention of methods to minimize bias (e.g., blinding of data evaluators) — this should be acknowledged.
Answer: Thank you for this important observation. We have now addressed this in both the materials and methods (2.7.Statistical Analysis). All data collection and educational sessions were conducted by one unblinded dietitian, which we acknowledge as a limitation. However, the statistical analysis was performed by a second author who was blinded to group allocation and had no contact with participants. This partial blinding at the data analysis stage helped reduce the risk of observer bias. Please see lines 159-162.
- Although results are statistically significant, the magnitude of changes (especially metabolic parameters like cholesterol or fasting glucose) is modest. Please discuss the clinical relevance of these findings.
Answer: Thank you for this valuable comment. In response, we expanded the discussion section to include a critical reflection on the clinical relevance of the observed metabolic changes. While parameters such as total cholesterol and fasting glucose showed statistically significant improvements, we acknowledged that the absolute magnitude of these changes was modest. We emphasized that these findings should be interpreted with caution and may be of potential importance primarily in the context of long-term dietary interventions. Please see lines 304-310.
- Multiple statistical comparisons were performed without correction for multiple testing (e.g., Bonferroni adjustment) — the risk of false positives should be acknowledged.
Answer: Thank you for this valuable remark. While we acknowledge the issue of multiple comparisons, the current analysis was primarily exploratory in nature, aimed at detecting potential signals of effect. Therefore, no formal correction (e.g., Bonferroni) was applied. However, to address the concern, we have emphasized in the Limitations of the study that the observed changes, although statistically significant, were modest in magnitude. Their clinical relevance remains limited but may still suggest favorable trends, especially in the context of a longer-term nutritional intervention. These results should thus be interpreted with caution and seen as preliminary, encouraging further targeted research. Please see lines 370-372.
- The discussion tends to overstate the impact of the intervention (particularly in asthma control), considering the limitations of subjective reporting and the lack of objective lung function tests. Please moderate the conclusions accordingly.
Answer: Thank you very much for your insightful and constructive comments. In response, we have revised the conclusions section to more cautiously reflect the observed outcomes, particularly with regard to asthma control. We have acknowledged the reliance on subjective reports and the absence of objective lung function assessments such as spirometry and emphasized that the results should be interpreted with caution. Please see lines 338-346.
- Limitations related to selection bias (non-randomized sample), reliance on self-reports, and heterogeneity of consultation mode (in-person vs. online) should be discussed more critically.
Answer: Thank You for this observation. The Limitations of the study section has been expanded to more clearly discuss the methodological constraints of the study, including the non-randomized sample, the use of self-reported data, and the heterogeneity in the mode of consultations (in-person vs. online). These revisions aim to provide a balanced interpretation of the findings and clarify the scope of the study. Please see the lines: 349-364.
- The English language can benefit from professional editing to improve flow and clarity. Many sentences are unnecessarily long and could be broken down for better comprehension. Minor translation oversights (e.g., "albo") should be corrected.
Answer: Thank You. Language editing has been done. - General Recommendations
Add a stronger paragraph on the limitations of the study, focusing on subjective measures, self-report biases, and the potential impact of online consultations.
Answer: Thank You. Please see lines 349-364.
Refine the conclusions to align more conservatively with the actual strength of the evidence presented.
Answer: Thank you for your comment. Done. Please see lines 338-346.
Perform a language review to enhance fluency, reduce redundancy, and correct minor grammatical errors.
Answer: Language editing has been done.
Optionally, consider presenting the education intervention process in a schematic table or flowchart to enhance clarity.
Answer: Thank you for the suggestion. A schematic table presenting the education intervention process has been added to improve clarity. Please see Table 1.
Reviewer 2 Report
Comments and Suggestions for Authors
In this manuscript the authors explore the effects of diet and nutrition education in a asthma and obesity pediatric population. The focus of the work is very interesting, and the manuscript is well organized, but there are some improvements to be made.
- The authors, both in the title and in the introduction, talk about diet and nutritional education, but in the methods, it is not clear that the subjects are subjected to a diet, explain.
- In the “Study Design and Ethics Issue” delete “This study was conducted in accordance with the principles of the Declaration of Helsinki”, line 94.
- In the methods there is not C-reactive protein, which is described in the results, add.
- Describe both the questionnaires and how the scores were obtained to assess motivation and knowledge.
- In the legend of table 4 move (AO-before).
- Lines 213-216 are very confusing, simplify and explain better.
- Line 241, specify which score it refers to.
- Explain the different representations of scores in Figures 2a and 2b and in Figures 3a and 3b, both in the text and in the legends.
- Did the authors analyze the data by sex? The authors should report the results by sex, specifying whether there were differences or not. If they did not analyze the data by sex, explain and report this in the limitations.
Author Response
Response to Reviewer 2:
Thank you for the time and effort in reviewing and providing feedback on our manuscript, and we are grateful for the insightful comments and valuable improvements to our paper. Below, we provide the point-by-point responses.
- The authors, both in the title and in the introduction, talk about diet and nutritional education, but in the methods, it is not clear that the subjects are subjected to a diet, explain.
Answer: Thank you for this important remark. We clarified in the revised methods section that participants did not follow a fixed diet plan. Instead, they received individualized dietary recommendations from a dietitian during consultations every 6–8 weeks. The guidance was based on the Mediterranean and anti-inflammatory dietary patterns and aimed to help families gradually improve dietary habits. This has now been clearly stated to better reflect the nature of the intervention. Please see lines 140-147.
- In the “Study Design and Ethics Issue” delete “This study was conducted in accordance with the principles of the Declaration of Helsinki”, line 94.
Answer: Done. Please see the lines: 78-79.
- In the methods there is not C-reactive protein, which is described in the results, add.
Answer: Added. Please see the line: 110.
- Describe both the questionnaires and how the scores were obtained to assess motivation and knowledge.
Answer: Thank you for this comment. As clarified in the revised methods section, both motivation and nutrition knowledge were assessed using a subjective tool — the Visual Analogue Scale (VAS). Participants marked their perceived level of motivation and knowledge on a 10-point horizontal line, with 0 indicating “very low” and 10 indicating “very high.” This method was used for its simplicity and suitability for pediatric populations.
- In the legend of table 4 move (AO-before).
Answer: Moved. Please see the legend of table 4.
- Lines 213-216 are very confusing, simplify and explain better.
Answer: Thank you for pointing this out. We have revised this paragraph to improve clarity and simplify the structure. We have removed redundancy, clarified the direction of changes in both groups, and included the information on statistical significance. The revised version now reads:
“After the intervention, a significant increase in the YHEI index was observed only in group O (p < 0.05). This group also had the highest YHEI scores at both baseline and post-intervention. In contrast, the YHEI index in the control group remained unchanged (Figure 1a).” Please see lines 213-217.
- Line 241, specify which score it refers to.
Answer: Clarified. Please see line 244.
- Explain the different representations of scores in Figures 2a and 2b and in Figures 3a and 3b, both in the text and in the legends.
Answer: We appreciate your valuable observation. According to the suggestion, the modification was done. Please see the lines: 239-241; 252-253; 256-258; 265-266.
- Did the authors analyze the data by sex? The authors should report the results by sex, specifying whether there were differences or not. If they did not analyze the data by sex, explain and report this in the limitations.
Answer: Thank you for this valuable suggestion. We acknowledge that sex-based analysis is an important aspect of clinical and nutritional research. In the present study, sex-specific differences were not analyzed due to insufficient statistical power when dividing the already stratified groups (O, AO, and C) further by sex. We have now clarified this point in the limitations of the study section of the manuscript. We also added a recommendation for future studies to include larger sample sizes that allow for meaningful subgroup analyses by sex. Please see lines 365-369.
Round 2
Reviewer 1 Report
Comments and Suggestions for Authors
Thanks you for your replies
Author Response
Thank you for the time and effort in reviewing and providing feedback on our manuscript.
Reviewer 2 Report
Comments and Suggestions for Authors
I thank the authors for accepting the suggestions, the manuscript is much improved.
Author Response

(The authors gave the same response as above.)
